# Non-Thermal Plasma Jet-Treated Medium Induces Selective Cytotoxicity against *Mycobacterium tuberculosis*-Infected Macrophages

**DOI:** 10.3390/biomedicines10061243

**Published:** 2022-05-26

**Authors:** Chae Bok Lee, Kang In Lee, Young Jae Kim, In Taek Jang, Sintayehu Kebede Gurmessa, Eun Ha Choi, Nagendra Kumar Kaushik, Hwa-Jung Kim

**Affiliations:** 1Department of Microbiology & Medical Science, College of Medicine, Chungnam National University, Daejeon 301-747, Korea; cblee2017@cnu.ac.kr (C.B.L.); popigleton@nate.com (K.I.L.); invivor@gmail.com (Y.J.K.); janginteak@naver.com (I.T.J.); sintayehu.kebedeg@gmail.com (S.K.G.); 2Plasma Bioscience Research Center, Department of Electrical and Biological Physics, Kwangwoon University, Seoul 01897, Korea; ehchoi@kw.ac.kr (E.H.C.); kaushik.nagendra@kw.ac.kr (N.K.K.)

**Keywords:** plasma-treated medium, tuberculosis, lesion, surgery, nonthermal plasma

## Abstract

Plasma-treated media (PTM) serve as an adjuvant therapy to postoperatively remove residual cancerous lesions. We speculated that PTM could selectively kill cells infected with *Mycobacterium tuberculosis* (*Mtb*) and remove postoperative residual tuberculous lesions. We therefore investigated the effects of a medium exposed to a non-thermal plasma jet on the suppression of intracellular *Mtb* replication, cell death, signaling, and selectivity. We propose that PTM elevates the levels of the detoxifying enzymes, glutathione peroxidase, catalase, and ataxia-telangiectasia mutated serine/threonine kinase and increases intracellular reactive oxygen species production in *Mtb*-infected cells. The bacterial load was significantly decreased in spleen and lung tissues and single-cell suspensions from mice intraperitoneally injected with PTM compared with saline and untreated medium. Therefore, PTM has the potential as a novel treatment that can eliminate residual *Mtb*-infected cells after infected tissues are surgically resected.

## 1. Introduction

Surgery used to be the sole treatment option for tuberculosis (TB), which has become a global epidemic. Although the advent of anti-TB drugs has helped 95% of patients with TB, surgery is still needed for ~5% of patients. However, surgery is hampered by the need to remove bacteria persisting inside tissues [1,2]. Plasma-treated media (PTM) represent a practical alternative for treating cancer lesions. Macrophage apoptosis is associated with increased reactive oxygen species (ROS) production [3]. However, direct plasma and PTM at atmospheric pressure induces ROS production in various cell types [4] and ROS-mediated cell death against bacterial infections [5]. Incremental ROS are significant as they are signaling molecules that play crucial roles in killing intracellular pathogens in macrophages [3,6]. An imbalance between ROS production and elimination leads to the initiation of cellular signaling factors that cause oxidative stress [7] in phagosomes of phagocytes containing pathogens; this is an important part of antimicrobial immunity [8,9].

Reactive oxygen and nitrogen species (RONS) generated by non-thermal plasma jet (NTPJ) were considered the main effector of oxidative stress [10]. Mediators associated with RONS also function in cancer, chronic wound healing, and chronic liver and lung diseases in humans [11,12]. Plasma-treated liquids or media act as transport media for plasma-generated species and comprise a branch of plasma medicine. Therapy with plasma-treated solutions is based on the effect of relatively long-lived ROS and reactive oxygen species (RNS) [13]. Plasma-treated saline can also sensitize colon cancer cells and modulate the antioxidant defense system [14]. Plasma-activated medium (PAM) leads to apoptosis in lung cells via spiral apoptotic cascade signaling that involves the mitochondrial-nuclear network [15]. Hydrogen peroxide (H_2_O_2_) among all reactive species present in PAM or PTM initiates a signaling cascade [16,17,18] that induces apoptotic factors in cancer cells [13,15,19]. Tornin et al. verified that PAM containing sodium pyruvate (NaPyr) significantly reduced the production of H_2_O_2_ and ROS but was still cytotoxic against osteosarcoma [20]. However, Li et al. found that neither low H_2_O_2_ nor high NO_2_^−^ and NO_2_^−^/NO_3_^−^ concentrations in PAM containing NaPyr mediate the anti-proliferative effects of PAM [21]. Our recent findings indicated that plasma-generated nitric oxide (PGNO)-media mainly contain reactive nitrogen species (RNS) that activate macrophages without affecting their cellular metabolic activity [22]. In addition, PAM suppresses gastric cancer cell motility, and migration and adhesion capacity, in vitro. The intraperitoneal administration of PAM inhibits peritoneal dissemination without side effects *in vivo* [23,24]. Despite these validated experimental findings of biological and medical applications of PTM [13], the effects of plasma-based radicals dissolved in liquids on Mycobacterium tuberculosis (*Mtb*)-infected macrophages have not yet been identified as potential therapeutic benefits of RONS. Pathogenic and drug-resistant *Mtb* strains have excellent sensitivity to increased endogenous ROS [25,26], indicating the importance of targeting intracellular and intra-mycobacterial redox metabolism for controlling TB infection. We previously found that NTPJ-based reactive species can effectively induce *Mtb* inactivation in aqueous solutions [27]. Water vapor introduced into a non-thermal plasma jet (NTPJ) device generates more hydroxyl radicals and consequently H_2_O_2_ as a long-lived species in liquids [28,29]. The generated H_2_O_2_ might also participate in mycobacterial inactivation. These findings suggested that if PTM could selectively induce the death of cells infected with *Mtb*, it would be useful as supportive therapy to remove such cells after surgery in patients who do not respond well to antibiotics. This study aimed to determine whether PTM could induce selective cytotoxicity in *Mtb*-infected and uninfected macrophages. We tested our hypothesis by investigating cellular metabolic activity and the redox balance, induced cell death, growth and counts of *Mtb* bacterial in infected cells, and physical and chemical changes in cytoplasmic mitochondria.

## 2. Materials and Methods

### 2.1. Mice and Cells

Tibias and femurs were isolated from 5–6-week-old C57BL/6 mice, then bone marrow-derived macrophages (BMDMs) were generated, cultured, and purified as previously described [30]. BMDMs were differentiated in a medium containing 25 ng/mL macrophage colony-stimulating factor (M-CSF; PeproTech, Rocky Hill, NJ, USA) for 4–6 days. The BMDMs and Raw 264.7 cells were cultured in Dulbecco’s modified eagle medium (DMEM) (Welgene, Daegu, Korea) containing 10% fetal bovine serum (FBS; Hyclone, Logan, UT, USA) and 1% antibiotics (Welgene).

### 2.2. Ethics Statement

The Institutional Research and Ethics Committee at Chungnam National University approved all animal experiments (approval number: 202003A-CNU-064) that complied with the relevant guidelines of the Korean Food and Drug Administration.

### 2.3. Non-Thermal Plasma Jet Device

Appendix A show a plasma jet device operated at atmospheric pressure that produces high concentrations of H_2_O_2_. Plasma is generated between electrodes without a porous ceramic-based dielectric barrier using 1.5 standard liters per minute (SLM) of N_2_ gas at atmospheric pressure [29]. A high voltage was supplied to the jet source by an AC commercial transformer. Waveforms were generated at a voltage of 1.06 kV (TDS2002C; Tektronix) and a current of 4.5 mA (P6021, Tektronix, Beaverton, Oregon) (Appendix A). Optical emission spectroscopy (OES) of plasma in the 200–500 nm range was recorded using an HR4000 spectrometer (Ocean Optics Inc., Dunedin, FL, USA). The distance between the plasma exit and the surface of the medium was fixed at 5 mm. We prepared PTM by applying an NTPJ to 500 µL/well of DMEM without (Sigma-Aldrich Corp., St. Louis, MO, USA) or with (Welgene, Taipei, China) NaPyr in 24-well culture plates for 5 min (Figure 1). The PTM was then diluted in a medium with or without NaPyr.

### 2.4. Assessment of H_2_O_2_ and NO_2_ Levels

We exposed DMEM with (DMEM) or without NaPyr (DMEM(-)NaPyr) (0.5 mL/24 well-plate) to NTPJ with N_2_ for 1 and 5 min (Figure 2B–D). The generation of H_2_O_2_ in the medium was colorimetrically analyzed using a working solution comprising 50 µL of 10 mM Amplex™ Red and 100 µL of 10 U/mL horseradish peroxidase (HRP; A22188, Invitrogen, Carlsbad, CA, USA) stock solution [31]. We applied NaPyr (11360-070, 100 mM, Life Technologies, Carlsbad, CA, USA) to scavenge peroxide. Nitrite in the media was analyzed using the Griess reagent, which comprises 0.1% sulfanilamide and 1% naphthylethylene dihydrochloride (Sigma Aldrich). The standard was nitrite (100 mM) diluted in distilled water and the NO scavenger was 2-(4-carboxyphenyl)-4,4,5,5-tetramethylimidazoline-1-oxyl-3-oxide, (cPTIO; Cat. No: 81540; Cayman Chemical Co., Ann Arbor, MI, USA). Medium containing 3% H_2_O_2_ was diluted and then also added to the media.

### 2.5. Assessment of ONOO^−^ Concentration Using LC-MS

Concentrations of 3-nitro-L-tyrosine (3-NT; N-7389, Sigma-Aldrich) were measured as an indicator of ONOO^-^ using LC-MS and a standard curve of 3-NT [32]. We measured ONOO^−^ by injecting 20 µL of solution into an Eclipse XD8-C18 system (4.6 × 150 mm; Agilent Technologies, Santa Clara, CA, USA), with a mobile phase consisting of an acetonitrile/1% formic acid (10:90 *v*/*v*) gradient from 10% to 90% at 25 °C. Samples were analyzed using a Sciex API 4000QTRAP mass spectrometry system (6120, Agilent Technologies) in ESI-positive ion mode under the following conditions: fragmentor, 70 V; drying gas flow, 12 L*·*min^−1^; drying gas temperature, 350 °C; nebulizer pressure, 35 psig; capillary voltage, 2.5 kV. The mass range scan in mode was m/z 50–200 kDa. The flow rate was 0.4 mL*·*min^−1^, and the retention time of 3-NT was 7.823 min. We detected 3-NT at 226.19. We calibrated 3-NT by measuring the retention time of 0, 3.125, 6.25, and 12.5 µM standard solutions (Appendix A). Before using plasma, 0.1 or 1 mM of L-tyrosine (T-3754, Sigma-Aldrich Corp.) in 0.5 mL of DMEM with or without NaPyr was incubated with DMEM in 0.1 M sodium phosphate buffer (pH 7.0) for 5 min at room temperature [33]. The medium containing 0.1 mM L-tyrosine was exposed to plasma for 5 min. These media were flash-frozen immediately thereafter.

### 2.6. Bacteria Strains and Culture

Virulent H37Rv (ATCC 27294) and avirulent H37Ra (ATCC 25177) *Mtb* strains (American Type Culture Collection, ATCC; Manassas, VA, USA) were grown in Middlebrook 7H9 medium supplemented with 10% (oleic acid, albumin, dextrose, and catalase (OADC; BD Biosciences, San Jose, CA, USA) and 0.2% glycerol for 21–28 days. We cultured an *Mtb* strain expressing enhanced red fluorescent protein (ERFP) in Middlebrook 7H9 medium supplemented with 10% OADC and 50 μg/mL kanamycin (Sigma-Aldrich Corp.) [34].

### 2.7. Infection of Cells

Macrophages were seeded into 96- or 24-well plates on the day before infection and incubated for 12 h. The cells were infected with *Mtb* for 4 h at a multiplicity of infection (MOI) of 1, then the medium was removed, and the cells were incubated with bacterial culture medium containing 200 µg/mL amikacin (Sigma Aldrich Corp.) for 2 h.

### 2.8. Analysis of Colony-Forming Units (CFU)

The infected cells were lysed in distilled water, serially diluted, and plated on 7H10 agar, and colony-forming units (CFUs) were measured after 2 or 3 weeks. The viable bacteria were calculated and plotted as mean CFU/mL from triplicate wells.

### 2.9. CellTiter-Glo^®^ Luminescence Assay

The generation of ATP was assessed using CellTiter-Glo^®^ Luminescence cell viability (G7572, Promega, Madison, WI, USA) as described by the manufacturer. Cells were exposed to PTM for 24 h, then incubated at room temperature for 10 min in reagent (media vs. reagent, 1:1). Luminescence was measured using a Synergy HT microplate reader (Biotek Instruments, Winooski, VT, USA).

### 2.10. CCK-8 Assay

Cell viability after exposure to plasma or incubation with other reagents was assessed using CCK-8 assays (CK04; Dojindo Laboratories Co. Ltd., Kumamoto, Japan). The cells were incubated in a medium containing 10% CCK-8 solution for 1–2 h in darkness, then absorbance was measured at 450 nm using a microplate reader (Biotek). Staurosporine (STS) was the positive control for cell death.

### 2.11. Annexin V/PI

Raw 264.7 cells (2 × 10^4^/12-well plate) and BMDMs (2 × 10^5^/12-well-plate) were stained using FITC Annexin V BV421 (Annexin; BD Biosciences, San Jose, CA, USA) and propidium iodide (PI; BD Biosciences). Cells were analyzed by gating using the antigen-presenting cell (APC) surface marker F4/80 (Invitrogen) with log-phase controls and a Fortessa™ flow cytometer (BD Biosciences) (Appendix A). Samples were acquired for 10,000~30,000 events. Data were analyzed and gated using FlowJo software v. 10 (Tree Star, Ashland, OR, USA).

### 2.12. Intracellular ROS Analysis

Raw264.7 cells (2 × 10^4^/12-well plate) or BMDMs (2 × 10^5^/12-well plate) were stained with 10 μM H2DCFDA (C6827, Life Technologies) for 30 min at 37 °C with CO_2_, in darkness. Stained cells were analyzed using the Fortessa™ flow cytometer. Fluorescence intensity was acquired for 20,000–50,000 events/sample and analyzed by gating using F4/80 and log-phase controls (Appendix A).

### 2.13. RNA Extraction and RT-PCR

Total RNA was extracted from the cells (1 × 10^5^/48-well plate) using TRIzol™ (15596-026, Thermo Fisher Scientific Inc., Waltham, MA, USA) and acid guanidinium thiocyanate-phenol-chloroform [34]. The concentration and purity of the RNA defined as the ratio of absorbance at 260 and 280 nm (A260/A280 nm) were measured using an ND-1000 NanoDrop™ spectrophotometer (Qiagen GmbH, Hilden, Germany). Complementary DNA (cDNA) was synthesized from total RNA using Cycle Script RT Premix (dT20, Bioneer, Daejeon, Korea). Sequences of interest were amplified by quantitative real-time PCR (RT-qPCR) using Rotor-Gene SYBR Green PCR Master Mix (204076, Qiagen GmbH) under 30 cycles of 95 °C for 30 s, 55 °C for 30 s, and 72 °C for 30 s [34]. Relative mRNA expression was calculated using the 2^−ΔΔct^ method [35] and normalized to that of 18S rRNA. Appendix A lists the primer sequences. The experiments were repeated at least three times.

### 2.14. Analysis of Plasma Membrane Integrity

BMDMs (1 × 10^5^) were stained with 5 µM SYTOx™ (I35102; Invitrogen) for 10 min at 37 °C and fixed overnight in 1% paraformaldehyde at 4 °C. Images were acquired using an SP8 confocal microscope (Leica Microsystems GmbH, Wetzlar, Germany) and argon laser with dual emissions at 561 nm for excitation of *Mtb*-enhanced ERFP and 488 nm to detect SYTOx™. The images were visualized using a 40× objective and assessed using LAS X v. 3.7 software (Leica Microsystems GmbH). Membrane damage in infected cells was determined by counting the number of pixels for SYTOx™ (green channel) per cell with and without bacterial cells. Data were statistically analyzed using Prism v. 5 (GraphPad Software Inc., San Diego, CA, USA).

### 2.15. Immunofluorescence Staining

We seeded BMDMs (1 × 10^5^/24-well plate) on 15-mm coverslips and then stained them with 250 nM MitoTracker™ Red (catalog no. CMXRos M7512; Invitrogen) for 30 min. The cells were then fixed with 4% paraformaldehyde for 10 min, washed with PBS, and permeabilized with 0.1% Triton™ X-100 for 10 min. Non-specific antigen binding was blocked with 3% bovine serum albumin in PBS with 0.1% Tween-20 (PBS-T) for 1 h. The coverslips were washed with PBS and incubated with anti-cytochrome c (Cyt c) diluted to 1:100 (catalog no. D18C7; Cell Signaling Technology, Danvers, MA, USA) or Bax antibody (catalog no. 2772S; 1:100; Cell Signaling) in 3% bovine serum albumin in PBS-T overnight at 4 °C. The cells were washed with PBS-T wash buffer and incubated with secondary antibody (1:200) for 1 h. Nuclei were stained with 5 µM/mL Hoechst 33,342 (catalog no. H1399; Invitrogen), then the coverslips were mounted (Dako, Glostrup, Denmark). Fluorescent images were analyzed using an SP8 confocal microscope (Leica Biosystems) equipped with a three-line sequential system comprising the UV-DAPI laser (7%) and argon 488 laser (14%) with a photomultiplier tube detector and a DPSS 561 laser (1%) with a hybrid detector. The images were processed using LAS X 3.7 (Leica Biosystems), and ratios (%) of co-localized Bax:Cyt c stained with MitoTracker (red) dye were calculated by dividing the number of co-localized cells. At least 20 cells were plotted for each condition and means (*p* < 0.05) were calculated using GraphPad Prism 9.

### 2.16. Mouse Infection Models

Female C57BL/6 mice (age, 5–6 weeks) were each infected intravenously (i.v.) with 1 × 10^7^ CFU *Mtb* H37Ra in 200 µL PBS. Two or three weeks after infection, the mice were randomly assigned to receive untreated media, saline, PTM without NaPyr, followed by an intraperitoneal (i.p.) injection of 200 µL of 4- or 8-fold-diluted PTM without NaPyr The lungs and spleens from the mice in each group were divided into four sections and incubated with 4-fold-diluted PTM without NaPyr for 1 h. The spleen and lung sections were homogenized and divided into untreated and PTM-treated groups and incubated for 1 h with 4-fold-diluted PTM without NaPyr. Bacteria were counted in 10-fold serial dilutions of individual whole spleens or lung homogenates. The Institutional Research and Ethics Committee at Chungnam National University approved the procedures for mouse infection, housing, euthanasia, and processing *Mtb*-infected tissues (Approval no: 202003A-CNU-064) and proceeded in Bio-Safety Level 2 (BSL2) containment facilities.

### 2.17. Statistical Analysis

Data were statistically analyzed using GraphPad Prism 6 (GraphPad Software Inc.). All results are expressed as means ± standard error of the means (SEM) of three independent experiments. Differences between two groups were evaluated using paired Student *t*-tests and those among multiple groups were evaluated by one-way analyses of variance (ANOVA) followed by appropriate multiple comparison tests.

## 3. Results

### 3.1. Physical Properties of N_2_ Gas NTPJ and Long-Lived Species in PTM with or without NaPyr

We initially identified various plasma species excited by NTPJ. Figure 2A shows the emission spectra in the NTPJ range of 200–500 nm during 1.5 SLM N_2_ gas injection. Highly reactive species were dominated by RNS that consisted of NO radicals, second and first positive band systems of N_2_ in the ranges of 200–280 and 300–390 nm, respectively [36], and ionized nitrogen molecules in the range of 390–480 nm. Abundant reactive species such as OH at 309.4 nm were detected due to contact with ambient air. The generation of H_2_O_2_ by NTPJ time-dependently increased and was significantly higher in DMEM without NaPyr than with it, at 5 min (492 ± 39 vs. 80 ± 10 µM; *p* < 0.001) (Figure 2B). Figure 2C and Appendix A show an indirect LC-MS analysis of ONOO^-^ using a standard curve of 3-NT as an indicator of peroxynitrite. The ONOO^-^ concentration after 5 min of NTPJ was 2.98 µM and undetectable in DMEM with NaPyr, respectively. Nitrite (NO_2_^−^) concentrations quantified using the Griess reagent were more significantly enhanced in DMEM with than without NaPyr (290.12 ± 19.23 vs. 150.87 ± 15.74 µM; *p* < 0.05) (Figure 2D). Therefore, PTM was prepared by treating 500 µL of DMEM with and without NaPyr with plasma containing N_2_ for 5 min.

### 3.2. Plasma Treated Medium Induces Selective Cell Death of Mtb-Infected Macrophages

We investigated cell viability in 2-, 4-, 8-, and 16-fold-diluted PTM with and without NaPyr by measuring intracellular ATP (Figure 3A,C) and mitochondrial activity (Figure 3B,D). Figure 3A shows that twofold-diluted PTM induced 50% cell death in *Mtb*-infected and uninfected cells. The viability of cells infected with *Mtb* were reduced in the cells incubated with four- and eightfold-diluted PTM compared with uninfected cells. However, mitochondrial activity was significantly reduced in *Mtb*-infected cells incubated with eightfold-diluted PTM (50.09 ± 2.77% vs. 97.15 ± 8.20%; *p* < 0.05), but not in uninfected cells compared with untreated controls (Figure 3B). Figure 3C,D confirm the viability of cells incubated in PTM without NaPyr. Levels of ATP were significantly lower in *Mtb*-infected than uninfected cells incubated with eightfold-diluted PTM without NaPyr (60.21 ± 2.38% vs. 92.56 ± 4.38; *p* < 0.05) (Figure 3C). Mitochondrial activity was also significantly lower in *Mtb*-infected than uninfected cells (40.21 ± 3.77% vs. 98 ± 7.50%, *p* < 0.01; Figure 3D). Accordingly, cells exposed to 16-fold-diluted PTM without NaPyr were also lower in *Mtb*-infected than uninfected cells (63.21 ± 4.77% vs. 104 ± 2.13%, *p* < 0.05). These data indicated that the 8- and 16-fold dilution was not toxic to uninfected cells, whereas more cell death was caused by PTM in DMEM without than with NaPyr, and susceptibility to the effects of PTM was higher in *Mtb*-infected than in uninfected cells. Therefore, we prepared PTM using DMEM without NaPyr for subsequent experiments.

### 3.3. Cytotoxic Effects of PTM-Diluted Medium with and without NaPyr

Sodium pyruvate is a peroxide scavenger [15,37]. Although previous data regarding PTM anticancer activity closely correlated with our present results [15,21], the role of PTM with or without a scavenger has not been appropriately addressed. To determine whether NaPyr interferes with the activity of PTM diluted in a culture medium, we compared the cell viability between diluted PTM with and without NaPyr using CCK-8 assays (Figure 4A–D). The increase in cell density (from 5 × 10^3^ to 2 × 10^4^) between days 1 and 3 verified the effects of PTM without NaPyr. In addition, PTM without NaPyr exerted notable cell-death effects on *Mtb*-infected cells (5 × 10^3^) (Figure 3B,D). Figure 4A,B show that the cell viability was 25.12 ± 3.1% and 35.35 ± 2.13% (*p* < 0.001 for both) after incubation with fourfold-diluted PTM without NaPyr prepared in media with NaPyr and eightfold PTM diluted in media without NaPyr, respectively. However, the viability of uninfected cells was not reduced. Moreover, we compared the viability of cells exposed to H_2_O_2_ and STS. Incubation with H_2_O_2_ in a medium with NaPyr did not reduce the numbers of uninfected and *Mtb*-infected cells, whereas the viability of *Mtb*-infected cells incubated in H_2_O_2_ in a medium without NaPyr was reduced by 32.03 ± 2.5% (*p* < 0.05) compared with untreated controls. The viability of uninfected cells was not reduced. Furthermore, STS did not alter cell viability. Figure 4C shows a similar trend after incubating *Mtb*-infected cells with fourfold-diluted PTM without NaPyr for 48 and 72 h, but uninfected cells were also affected (15.21 ± 0.31% vs. untreated control, *p* < 0.05). Incubation in culture media with NaPyr caused the selective toxicity between the two groups to disappear. The viability of *Mtb*-infected cells incubated with eightfold-diluted PTM in a culture medium without NaPyr was decreased by 30.23 ± 3.15% compared to that of the untreated control (*p* < 0.001; Figure 4D). Cytotoxicity against uninfected cells incubated with PTM without NaPyr decreased to 5% compared to that of the untreated control. This selectivity was also evident after incubation for 72 h (*p* < 0.001). These data implied that the media containing NaPyr reduced the cytotoxic effect of PTM and even slightly increased the dilution factor. Therefore, the medium without NaPyr was exposed to plasma and diluted with NaPyr-free media in further experiments. The cytotoxicity of PTM without NaPyr was confirmed by annexin and PI staining after 24 h of incubation. The annexin-positive cell population in *Mtb*-infected cells was decreased by 30.13 ± 10.23% (*p* < 0.05) after incubation for 24 h in eightfold-diluted PTM without NaPyr (Figure 4E and Appendix A). We measured the relative amount of intracellular ROS in macrophages after incubation for 6 h in PTM without NaPyr. Figure 4F and Appendix A show that intracellular ROS levels were 100% higher in *Mtb*-infected cells than in untreated cells, and the levels of the *Mtb* control or cells treated with PTM alone were similar to those of the control. However, ROS production was inhibited in the presence of N-acetylcysteine (NAC). 

### 3.4. Plasma and PTM Treatment Inhibit Intracellular Mtb Growth

We previously reported that the efficiency of PTM depends on the density or volume of cancer cells [15,38]. We investigated whether these parameters affected the selective toxicity of PTM in uninfected and *Mtb*-infected BMDMs. The viability of uninfected cells (5 × 10^3^) and 1 × 10^4^ cells incubated with fourfold-diluted PTM without NaPyr was decreased to 50 ± 5% (*p* < 0.01 for both). In contrast, PTM without NaPyr did not decrease the viability of 2 × 10^4^ uninfected cells (Figure 5A). The viability of 5 × 10^3^ and 1 × 10^4^
*Mtb*-infected cells incubated with fourfold-diluted PTM without NaPyr decreased to 70.12 ± 5.04% and 60.17 ± 5.13% (*p* < 0.01), respectively, whereas the effect was slight at a density of 2 × 10^4^ cells (Figure 5B). Trends in the viability of *Mtb*-infected cells incubated with 8- and fourfold-diluted PTM without NaPyr were similar but those in uninfected cells were not. Therefore, we applied the optimal dilution of PTM in subsequent experiments to ensure that only *Mtb*-infected cells were destroyed.

Direct plasma treatment induces morphological changes in macrophages. Figure 5C shows phase-contrast bright-field images of *Mtb*-infected cells exposed to an NTPJ for 0, 1, 2, and 5 min, and 24 h. The morphological features changed over time to relatively small and round cells compared with untreated *Mtb*-infected cells. However, the morphological features of *Mtb*-infected cells after 1 min of NTPJ were similar to those of untreated cells. One minute of exposure to the NTPJ did not destroy any BMDMs. Figure 5D shows cell death after exposure to NTPJ for 30, 60, and 90 s. After 60 s of the NTPJ, the viability of *Mtb*-infected cells was reduced by 70.12 ± 3.07% compared with the untreated control (*p* < 0.001), but that of uninfected cells was not. These results indicated that direct plasma application for 60 s resulted in selective toxicity. However, this disappeared after 90 s of NTPJ exposure between the two groups, whereas the viability of *Mtb*-infected cells incubated with NaPyr and media with NaPyr recovered. These data indicate that 60 s of NTPJ exposure did not reduce the viability of uninfected cells.

Based on these results, we determined the growth rate of *Mtb* in BMDMs (1 × 10^4^/well) exposed to NTPJ or PTM in culture media with or without NaPyr. Figure 5E shows that the growth of intracellular *Mtb* cells was significantly suppressed after 24 h of NTPJ or PTM without NaPyr (*p* < 0.001) compared with untreated cells. At 72 h, *Mtb* growth increased in the cells treated with NTPJ for 60 s compared with PTM without NaPyr. This indicated that PTM without NaPyr induced significantly more growth inhibition (*p* < 0.001) than NTPJ (*p* < 0.01). Incubation with PTM diluted in media with NaPyr suppressed the growth of intracellular *Mtb* after 24 h (*p* < 0.01) but had less of an effect at 72 h. Consistent with our previous results, intracellular *Mtb* replication was significantly inhibited after PTM without NaPyr treatment in culture media without NaPyr. 

### 3.5. Plasma Treated Medium Elevated Antioxidant-Detoxifying Enzyme Activity in Mtb-Infected BMDMs

Antioxidant-detoxifying enzymes control levels of intracellular ROS and RNS, which allows intracellular bacterial pathogens to survive an oxidative burst [39]. We quantified mRNA using RT-PCR to determine which signals lead to the enzymatic detoxification of RONS induced by PTM in macrophages at the transcription level (Figure 6A–C). The increase in ATM levels in *Mtb*-infected cells incubated with PTM is associated with DNA repair and recombination [40]. Levels of ATM were increased 25-fold in *Mtb*-infected cells incubated with 8-fold-diluted PTM without NaPyr, untreated *Mtb*-infected cells incubated with 1.5-fold-diluted PTM without NaPyr, and cells exposed to NTPJ for 60 s and incubated in 7-fold-diluted PTM without NaPyr. Incubation with 8-fold-diluted PTM without NaPyr increased glutathione peroxidase (GPx1) levels 5-, 1.8-, and 2-fold in *Mtb*-infected, uninfected, and untreated *Mtb*-infected cells, respectively (Figure 6A). Furthermore, GPx1 activity was reduced in a medium containing NaPyr after 60 s of the NTPJ and 250 µM H_2_O_2_ (Figure 6B). Catalase (CAT) levels were significantly increased by 5-, 3-, 4-, and 3.8-fold in PTM treated infected cells, PTM treated uninfected cells, untreated infected cells, and cells exposed to NTPJ for 60 s, respectively. However, CAT levels decreased in the culture medium containing NaPyr (Figure 6C). Physical plasma-based ROS and RNS that affect redox biology might be dynamic signaling molecules that disrupt the antioxidant system. We measured ATP luminescence-based viability after incubating cells with the antioxidants, NaPyr, NAC, and cPTIO, 24 h after incubation with PTM without NaPyr (Figure 6D). Both NaPyr and NAC similarly recovered cytotoxicity, whereas PTM decreased the viability of cells incubated with cPTIO, which was contrary to the effects of NaPyr or NAC. Cytotoxicity mediated by PTM was not affected by the NO scavenger, cPTIO. These findings suggested that hydrogen peroxide initiates a signal protein [16,17,18], which induces cell death in *Mtb*-infected cells.

### 3.6. Plasma-Treated Medium Renders Infected Cells Permiability

Direct exposure to plasma increases cell permeability [41] and indirect plasma exposure enhances cell membrane permeability [42]. We measured membrane permeability using SYTOx™ (Figure 7). The intracellular emission of fluorescent SYTOx™ was significantly more intense in *Mtb*-infected cells incubated with PTM without NaPyr than in cells infected with *Mtb* or incubated with PTM (Figure 7A). Moreover, fluorescence emission was significantly more intense in *Mtb*-infected than in uninfected cells after incubation for 48 h in PTM. We also measured the quantification of ERFP *Mtb* and SYTOx™ association in single cells (Figure 7B). The fluorescence intensity emitted by SYTOx™ increased in *Mtb*-infected cells incubated with PTM. 

### 3.7. Changes in Cyt c and Bax Localization in Mtb-Infected Cells Induced by PTM

Macrophages incubated in PTM without NaPyr developed an apoptotic profile of shrinkage and rounding that was similar to that in positive control cells (Appendix A). Therefore, we analyzed changes in the localization of Cyt c and Bax in *Mtb*-infected cells incubated with PTM without NaPyr using fluorescence microscopy. Figure 8A shows significantly decreased Cyt c in the mitochondria of *Mtb*-infected cells incubated with PTM without NaPyr vs. cells incubated with PTM only or control medium (Appendix A). Significantly more Bax was located in mitochondria and *Mtb*-infected cells incubated with PTM without NaPyr than in those incubated with PTM and *Mtb*-infected cells (Figure 8B and Appendix A). Figure 8C,D confirm these results by the findings of Cyt c or Bax co-localization in mitochondria. These data suggest that the apoptotic pathway is involved in PTM-mediated cell death.

### 3.8. PTM Suppresses Mtb Growth In Vivo

We investigated whether PTM could inhibit *Mtb* growth in a mouse model to determine potential clinical implications. The mice were infected with *Mtb* H37Ra, as shown in Figure 9A. Two weeks after the infection, we injected PTM to mice once daily for 2 days, then bacterial loads in excised lungs and spleens were measured three days later. The bacterial loads were significantly lower in the lungs of mice injected i.p. with fourfold-diluted PTM without NaPyr than in untreated mice injected with medium or saline (Figure 9B). Bacterial growth in the spleens of mice injected with four- or eightfold-diluted PTM without NaPyr was significantly decreased (*p* < 0.01 and *p* < 0.05, respectively) compared with the untreated mice injected with medium. However, the statistical differences between mice injected with saline and those injected with PTM without NaPyr decreased from *p* < 0.01 to *p* < 0.05. Two weeks after the infection, spleens or lungs excised from *Mtb*-infected mice that were not injected with PTM without NaPyr were divided into four sections and incubated with PTM without NaPyr (Figure 9C). Bacterial counts were lower in the lung tissues incubated with PTM than in those incubated with untreated medium (*p* < 0.01) (Figure 9D). The growth of *Mtb* was significantly inhibited in spleen tissues incubated with PTM without NaPyr (*p* < 0.001) compared with those incubated in untreated medium (Figure 9D). We cultured single-cell suspensions of tissues excised from mice (Figure 9C,E,F) at different cell densities in PTM without NaPyr to increase uniform effect of PTM and permeability of cells. The bacterial loads in the lung and spleen were measured three weeks after the infection. Fourfold-diluted PTM without NaPyr significantly inhibited *Mtb* growth (*p* < 0.001) in the lung and spleen single cell suspensions (Figure 9E,F). These results suggested that direct contact between PTM without NaPyr and cells would result in optimal antibacterial effects. Therefore, PTM could be used to remove residual *Mtb*-infected cells after the surgical resection of infected tissues.

## 4. Discussion

The H_2_O_2_ and ONOO^-^ generated by NTPJ or PTM are of great interest to plasma medicine researchers because these cause the selective induction of cancer inhibition factors [15,19]. However, to our knowledge, selective toxicity studies on the effects of direct and indirect plasma treatment have not been carried out for virulent *Mtb*-infected macrophages or organs. NTP-based selectivity on cancer can be explained in terms of intrinsic differences between cancer and normal cells [43,44,45]. Endogenous cellular ROS levels are higher in cancer, resulting in a higher baseline ROS concentration, thus making it easier for the NTP-generated species to reach the apoptosis-inducing threshold [11,46]. Indeed, normal cells contain higher levels of anti-oxidant enzymes that limit cellular toxicity by detoxifying ROS [46]. Usually, H_2_O_2_ diffuses into the transmembrane using the aquaporin channel on cytoplasmic membranes. Cancers express more aquaporin on their cell membranes, inducing faster uptake speed of H_2_O_2_ than normal cells [47]. Hence, the supply of H_2_O_2_ from PTM causes a specific biological pathway in cancer or diseased cells. 

In this study, we prepared a stimulated media using NTPJ to generate high H_2_O_2_ and ONOO^-^ concentration, and then examined the selective cell death in *Mtb*-infected cells or *in vivo* experiments. The results of the present study, as well as those of other reports, have found that cell culture media prepared in the presence of NaPyr result in decreased production of H_2_O_2_ and cell death [20,21]. Cortazar et al. found that RONS induced by PAM delivers on a living tissue with negligible damage on healthy cells. In contrast, their PAM inactivated SARS-CoV-2 and PR8 H1N1 viruses effectively [48]. It was reported that the synergistic effect of H_2_O_2_, RONS and a low pH may induce cell death of infected cells. Plasma-activated water (PAW) can effectively inhibit SARS-CoV-2 pseudovirus inactivation by damaging S-protein [49]. Additionally, PAW efficiently inactivates bacteriophages T4, Φ174, and MS2 through reactive species generated by plasma at atmospheric pressure [50]. These generated reactive species damaged both nucleic acids and proteins, demonstrating that singlet oxygen may have played a primary role in the inactivation. Recently, the plasma-activated oil (PAO) applied to infected wound and confirmed that the wound infection area was significantly reduced compared the untreated control [51]. It is claimed that carboxylic acid is produced when the C = C bond is broken by RONS in PAO through acidification reaction and showed antimicrobial activity. Most of the biological effect of PAM has been attributed to long-lived species such as H_2_O_2_, ONOO^−^, NO_2_^−^, NO_3_^−^, and organic radicals produced in the liquid phase after non-thermal plasma (NTP) exposure [10].

We observed that PTM prepared in the presence of NaPyr does not show a reduction in the ATP luminescence level (Figure 3A,C), while on the other hand, assessment of cell viability using CCK-8 assay showed that the selective toxicity was further slightly enhanced by PTM in the absence of NaPyr (Figure 3B,D). In addition, we observed that PTM prepared in the absence of NaPyr causes an increase in selective cell death, as detected by means of measurement of ATP level and cell viability assay. We investigated an approach of two dilution methods, diluted PTM with or without NaPyr, for treatments (Figure 4A,B). *Mtb*-infected cells exposed to PTM diluted in media with the presence of NaPyr induced cell death but did not show a persistent selective cytotoxic effect between uninfected cells and *Mtb*-infected cells (Figure 4C). In this study, cells exposed to PTM (-)NaPyr showed significantly enhanced cell death, with a prolonged selective effect in the absence of NaPyr (Figure 4D). NaPyr, an end product of glycolysis that plays a role in tricarboxylic acid (TCA) cycle, is known to serve as a scavenger of peroxide in aerobic oxidation [52] as well as non-enzymatic scavenging of ROS, including H_2_O_2_, resulting in the generation of acetate, water, and carbon dioxide. A recent report showed that NaPyr added to cell culture media inhibits the immune signaling but not influenza viral replication [52]. Reports revealed that *Mtb* grew faster in a medium with NaPyr as the sole carbon source than when glucose was added [53]. These findings indicated that NaPyr is the preferred carbon source of *Mtb*. *Mtb* is intracellular bacteria that can ingest NaPyr from host cellular resources and uses it as an energy source for survival and replication. Thus, exogenous NaPyr may be critical to determining substrate for energy production in intracellular *Mtb* (Figure 5E). We used scavengers such as NaPyr, NAC, and cPTIO to comprehend the components responsible for the selective toxic effects of PTM in *Mtb*-infected cells (Figure 6D). Our results suggested that nitrite (NO_2_^-^)-related RNS is not an important factor for this biological effect.

As shown in the results, the strong selective toxicity supports previous reports, which have shown that H_2_O_2_ is mainly responsible for apoptosis [15,31]. However, it must be further investigated to validate the role of each species in *Mtb* inactivation. As *Mtb* infection in macrophages, the macrophages activate an indispensable cascade signaling, including DNA repair response or apoptosis pathway [54]. ATM gene is a driver of DNA repair and apoptosis. Our results also showed that PTM treatment enhanced the ATM level in *Mtb*-infected cells, leading to significant cell death compared to different treatment (Figure 6A). PTM treatment also induces enhanced H_2_O_2_-linked GPx1 (glutathione:hydrogen-peroxide oxidoreductase) antioxidant enzymatic activity (Figure 6B). GP × 1 inactivates H_2_O_2_ in a reaction that oxidizes GSH (glutathione) to its disulfide form glutathione disulfide (GSSG) [55]. A high level of GPx1 and CAT was observed after 24 h of PTM(-)NaPyr treatment to *Mtb*-infected cells, unlike that in the case of PTM-medium containing NaPyr. Both GPx1 and CAT are regulated in the cellular response against oxidative stress in *Mtb*-infected cells treated with PTM (Figure 6B,C). Conclusively, diluted PTM without NaPyr may be a valuable tool to study the mechanisms of selective cell death on *Mtb*-infected cells. Previous reports showed that the cytotoxic effect of PAM in cell lines varied with cell density or its volume [15,38].

This density-dependent phenomenon was also observed in this study. There was a marked difference in selective toxicity between uninfected cells and *Mtb*-infected cells, according to cell number density (Figure 5A,B), and these ex vivo results supported this selective phenomenon at specific cell densities (such as 1 × 10^4^ cells/96-well-plate). The *Mtb*-infected cells exposed to eightfold-diluted PTM(-)NaPyr solution markedly suppressed the replication of intracellular *Mtb* (Figure 5E). However, for the *in vivo* experiments, only fourfold dilution was used, as the living mouse has several different influencing factors and a complex environment including body fluids, tissues, and cells.

Before the antibiotics era, surgery was the only option for TB treatment [1]. However, nowadays, surgery is only used for the treatment of complicated cases of pulmonary TB, especially patients with drug-resistant tuberculosis, who do not respond to clinical drug treatment. Plasma can serve as an advanced approach to treat TB. However, the treatment of endogenous parts or organs deep inside the body using direct plasma is difficult. Therefore, PTM or plasma-treated liquids may be an efficient treatment method that can be used as a future therapeutic agent to treat TB. PTM has been shown to have a potential effect on cancer tumors *in vivo*, when subcutaneously injected at a volume of 100 µL into the tumor every other day [56]. The selectivity of PTM on tumors has been reported in both in vitro [43] and *in vivo* studies, without observable side effects on normal cells [56,57]. In line with previous studies, our results claim that PTM is more selective on *Mtb*-infected cells than uninfected cells ex vivo. It also suggests that cell death in *Mtb*-infected cells is mediated through Bax/Cyt c signaling (Figure 8 and Appendix A).

Here, we showed that PTM or plasma-based substances could eliminate postoperative residual TB lesions. A combination of PTM-based treatment and surgery would deliver optimal results. We optimized the doses of PTM and TB therapeutic agents. We found that eightfold-diluted PTM significantly reduced bacterial growth in the spleen compared with the control medium or saline, but not in the lungs *in vivo*. Our main focus was not to determine the direct activity of PTM injection. Instead, we further focused on PTM activity in the lungs and spleens of mice infected with *Mtb in vivo* (Figure 9C–F). We considered that this reflected the status of PTM treatment after infected tissues have been surgically excised. Excised lungs and spleens were cut into four sections (Figure 9D), separated into single cells (Figure 9E,F), and incubated with PTM. The potential clinical aims of this strategy using PTM were to decrease the cell density and allow for a more efficient use of antibiotics to treat infected cells [58] (Figure 9E,F). Our results notably suggested that single-cell suspensions of lung tissues resulted in increased intracellular PTM penetration (Figure 9E,F) compared with lung tissue sections (Figure 9D). Particularly, PTM without NaPyr might be more effective in single lung cells due to direct contact. However, we used different bacterial strains in the macrophage and animal models. The *Mtb* H37Rv and H37Ra strains are virulent and avirulent, respectively. Both are derived from the same parental strain but differ in terms of virulence in experimental animals. The avirulent H37Ra strain is easier to use experimentally because the risk of infection is avoided. We previously showed that H37Ra can be used to test vaccine efficacy in mouse models because H37Ra persists in mouse tissues for >26 weeks after infection [59]. We plan to analyze the effects of PTM in *Mtb* H37Rv in future studies.

Other studies have suggested that killing bacteria with antibiotics is density dependent. However, high antibiotic concentrations do not alter the degree of cell death for defined periods [58]. We showed that antibacterial activity at a low bacterial density did not significantly differ between the separate sections and single-cell suspensions of spleens (Figure 9D–F). These results are in line with previous findings. We showed that NTP-based PTM technology could be applied to remove residual *Mtb* after the surgical resection of infective tissues using a schedule of short-term and repetitive treatment strategies.

## 5. Conclusions

We illustrated that PTM induced selective cell death in *Mtb*-infected macrophages. Diluted PTM in NaPyr-free media resulted in efficient H_2_O_2_-linked Gpx1, CAT, and ATM activity, as well as intracellular ROS production. This disrupted the cell membrane and induced the activation of Bax/Cyt c signaling in *Mtb*-infected macrophages, which eventually led to the inhibition of *Mtb* growth. The effect of PTM on single-cell suspensions or tissues of lungs and spleens excised from infected mice increased compared with that of the untreated medium. Therefore, the present findings can be used as a basis for the potential application of PTM to treating TB and eliminating postoperative residual bacteria in patients with TB.

## Figures and Tables

**Figure 1 biomedicines-10-01243-f001:**
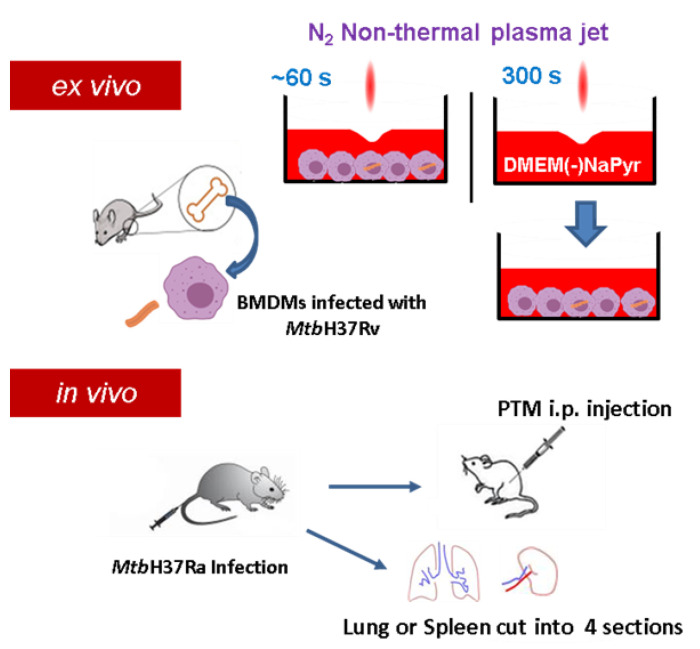
Scheme of non-thermal plasma jet to treat infected or uninfected cells *ex vivo* and *in vivo*. Cell cultures were exposed to a non-thermal plasma jet (NTPJ) ex vivo by placing the plasma exit 5 mm above medium surface for 30–60 s. Medium (DMEM) was exposed to NTPJ for 300 s to generate PTM. First mice were infected with *Mtb* H37Ra and after two or three weeks, the mice were injected intraperitoneally (i.p.) with PTM. Additionally, the spleens and lungs were obtained from the infected mice, cut into four pieces, and suspended into single cells, before exposure to PTM.

**Figure 2 biomedicines-10-01243-f002:**
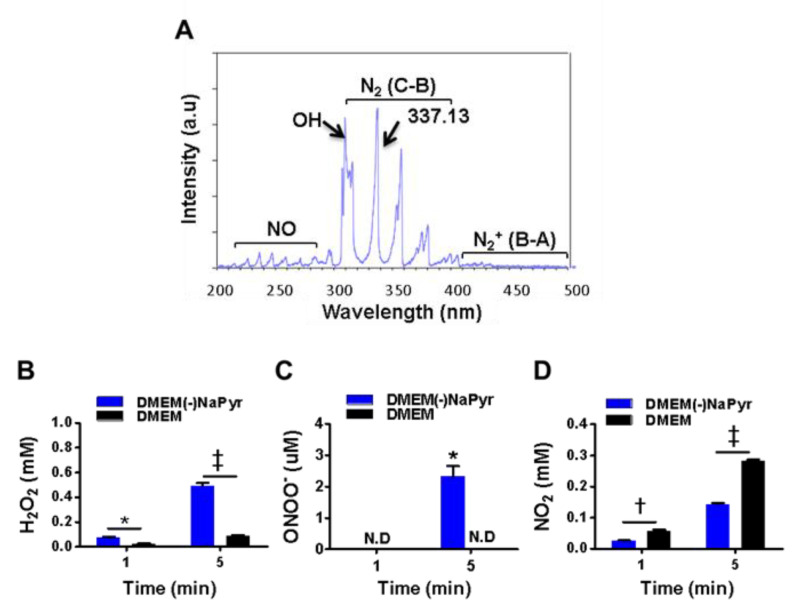
Optical emission spectra (OES) of a plasma jet with N_2_ gas and H_2_O_2_, ONOO^−^ and NO_2_ in two concentrations of DMEM with or without NaPyr exposed to plasma. (**A**) Optical emission spectra of plasma jet at 200–500 nm. (**B**–**D**) Assessment of H_2_O_2_, ONOO^−^, and nitrite (NO_2_^−^) concentrations in DMEM with or without NaPyr exposed to NTPJ for 1 and 5 min. N.D: not detected; * *p* < 0.05, and † *p* < 0.01, and ‡ *p* < 0.001, DMEM vs. DMEM(-)NaPyr. All error bars represent the mean ± standard deviation; n = 3.

**Figure 3 biomedicines-10-01243-f003:**
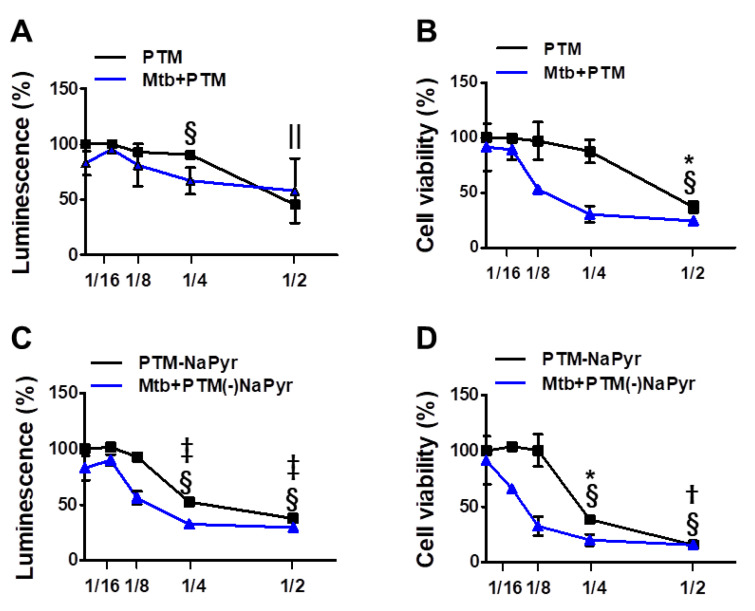
Assessment of ATP-based luminescent cell viability and mitochondrial activity of *Mtb*-infected (MOI 1) and uninfected cells exposed to PTM at different dilution ratios. (**A**–**D**) Dose responses of BMDMs to PTM determined using ATP luminescence (**A**,**C**) and CCK-8 (**B**,**D**) assays. Cells (5 × 10^3^/well in 96-well plates) were seeded overnight before incubation with PTM. Results are expressed as ratios of ATP-based viable and mitochondrial activities. (**A**,**B**) PTM, (**C**,**D**) PTM without NaPyr. PTM or PTM without NaPyr was diluted in media with NaPyr. Data are means ± SEM of at least three independent experiments. Statistical significance was determined for each dose compared with untreated control. (**A**–**D**) Uninfected cells: * *p* < 0.05, † *p* < 0.01, ‡ *p* < 0.001; *Mtb*-infected cells: ^§^ *p* < 0.05, and ^||^
*p* < 0.001; uninfected vs. infected cells * *p* < 0.05 and † *p* < 0.01.

**Figure 4 biomedicines-10-01243-f004:**
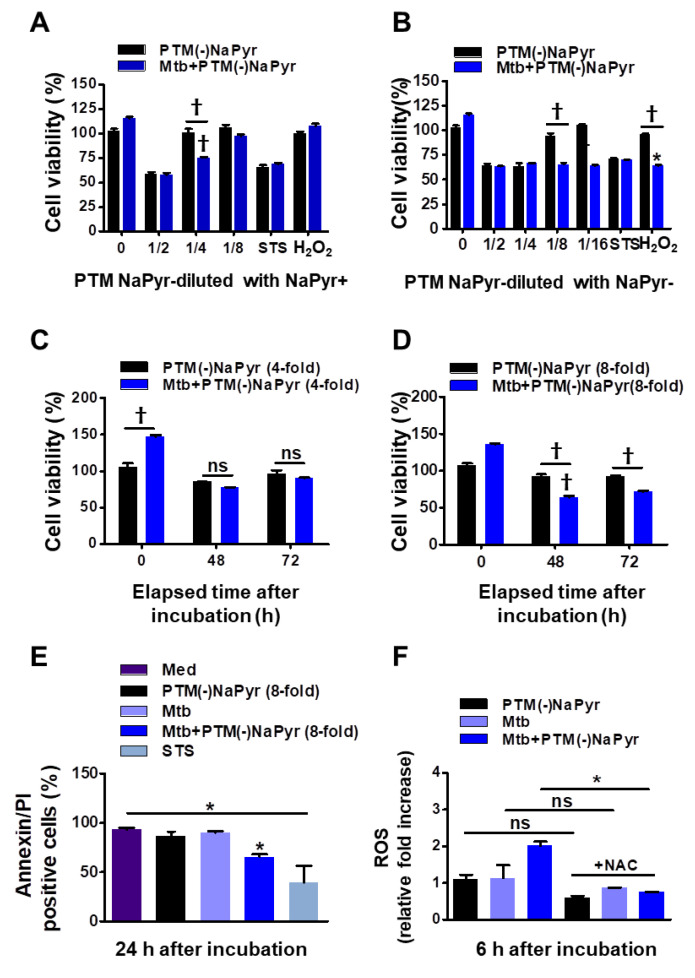
Comparison of effects of PTM without NaPyr diluted in medium with or without NaPyr. Uninfected or *Mtb*-infected BMDMs (2 × 10^4^/well of 96-well plates) (MOI 1) were incubated for 24 h in various dilutions of PTM without NaPyr in culture media with (**A**) and without (**B**) NaPyr, then cell viability was determined using CCK-8 assays. Effects of 250 µM H_2_O_2_ and 50 nM STS cell viability were also analyzed. Cells were incubated for 48 h and 72 h with (**C**) 4-fold-diluted PTM in medium with NaPyr and (**D**) 8-fold-diluted PTM in medium without NaPyr. (**E**) Death of Raw264.7 cells incubated in PTM without NaPyr evaluated using flow cytometry and annexin/PI staining. (**F**) Intracellular ROS in Raw264.7 cells incubated for 6 h in 8-fold-diluted PTM without NaPyr or PTM treated with NAC (10 mM) without NaPyr, and uninfected, infected control, and *Mtb*-infected cells incubated in PTM without NaPyr. (**A**–**F**) * *p* < 0.05 and † *p* < 0.001 vs. untreated cells; one- or two-way ANOVA. ns: not significant. Data are shown as means ± SEM of three independent experiments.

**Figure 5 biomedicines-10-01243-f005:**
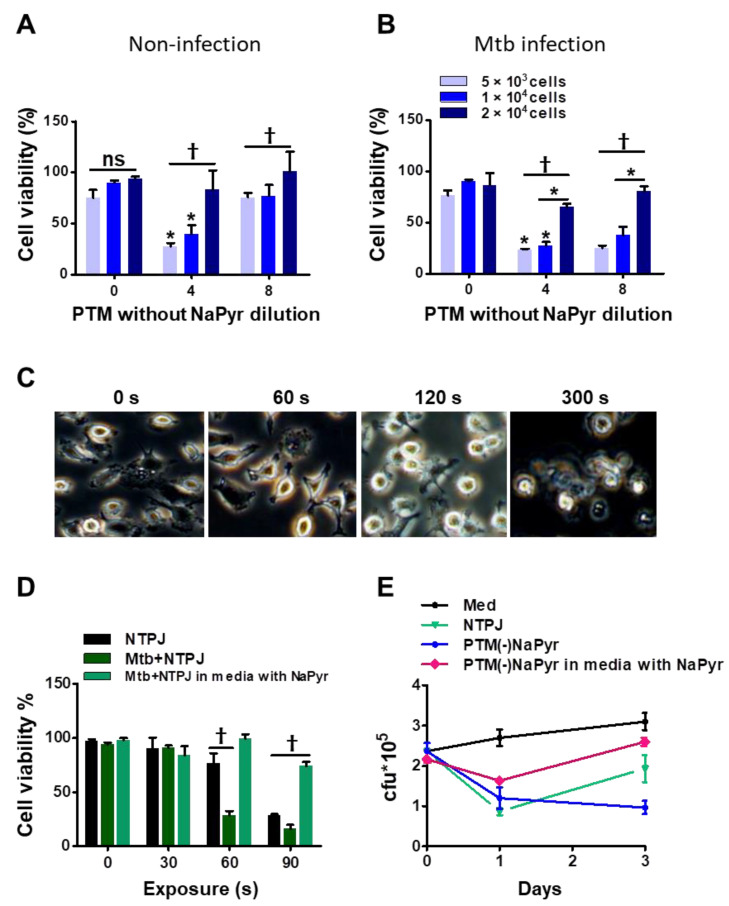
Identification of optimal PTM dilution to induce cell death ex vivo but remain non-toxic to uninfected BMDMs. Direct plasma NTPJ and PTM reduced intracellular *Mtb* growth. (**B**) Viability of (**A**) uninfected and (**B**) *Mtb*-infected cells (5 × 10^3^, 1 × 10^4^, and 2 × 10^4^/well of 96-well plates) determined 24 h after incubation with diluted PTM without NaPyr using CCK-8 assays. (**C**) Representative morphological features of *Mtb*-infected BMDMs after direct plasma exposure for 0, 60, 120, and 300 s. (**D**) Viability of uninfected and *Mtb*-infected BMDMs (1 × 10^4^/well of 96-well plates) after exposure to NTPJ for 30, 60, and 90 s in medium containing NaPyr under same conditions. * *p* < 0.01 and † *p* < 0.001; two- or one-way ANOVA. Error bars show the standard deviation of the means of three independent experiments. (**E**) Growth of intracellular bacteria assessed using CFU assays, immediately or one or three days after exposure to NTPJ for 60 s and 8-fold-diluted PTM in a medium with or without NaPyr. Colony-forming units were determined in cultures on 7H10 agar plates. *p* < 0.01, NTPJ and *p* < 0.001, PTM without NaPyr vs. untreated control on day 3; n = 3. *p* < 0.001, NTPJ and PTM without NaPyr vs. untreated control on day 1.

**Figure 6 biomedicines-10-01243-f006:**
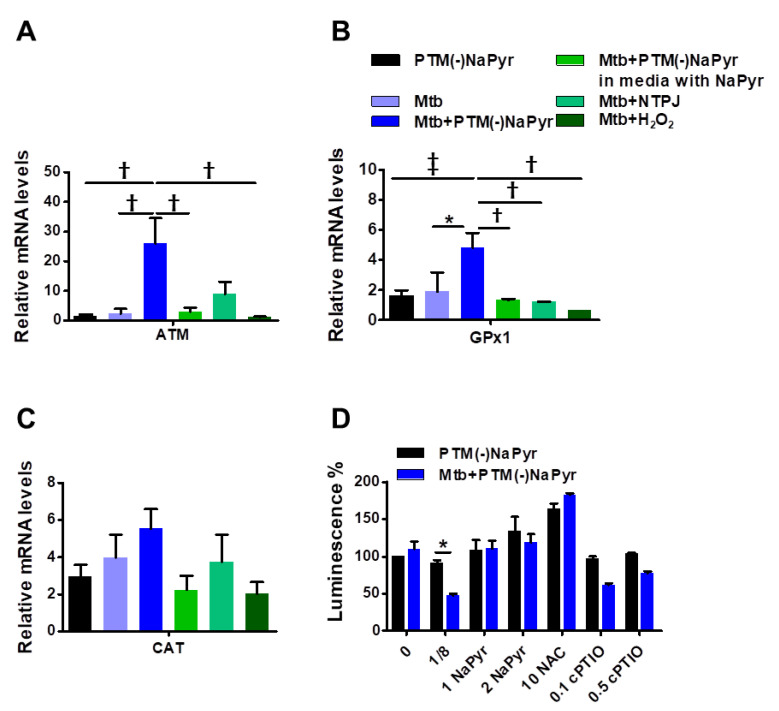
Plasma-treated medium stimulates antimycobacterial defenses in BMDMs. (**A**–**C**) Uninfected or *Mtb*-infected BMDMs (MOI 1) were incubated in undiluted and 8-fold-diluted PTM without NaPyr, then exposed for 60 s to NTPJ, and 250 µM H_2_O_2_ for 24 h in media without.NaPyr. *Mtb*-infected cells were incubated with NaPyr followed by 8-fold-diluted PTM without NaPyr) in medium containing NaPyr. The mRNA expression of ATM (**A**), GPx1 (**B**), and CAT (**C**) was quantified using RT-PCR. Fold induction was calculated by comparison with 18 S rRNA untreated control. (**D**) Control, uninfected, and *Mtb*-infected BMDMs were incubated with 8-fold-diluted PTM without NaPyr, 8-fold-diluted PTM with 1 and 2 mM NaPyr, NAC (10 mM), and 0.1 and 0.5 mM cPTIO for 24 h. Cell viability was assayed using the CellTiter-Glo^®^ assay. Error bars show standard deviations of the means. * *p* < 0.05, † *p* < 0.01, and ‡ *p* < 0.001; n = 3.

**Figure 7 biomedicines-10-01243-f007:**
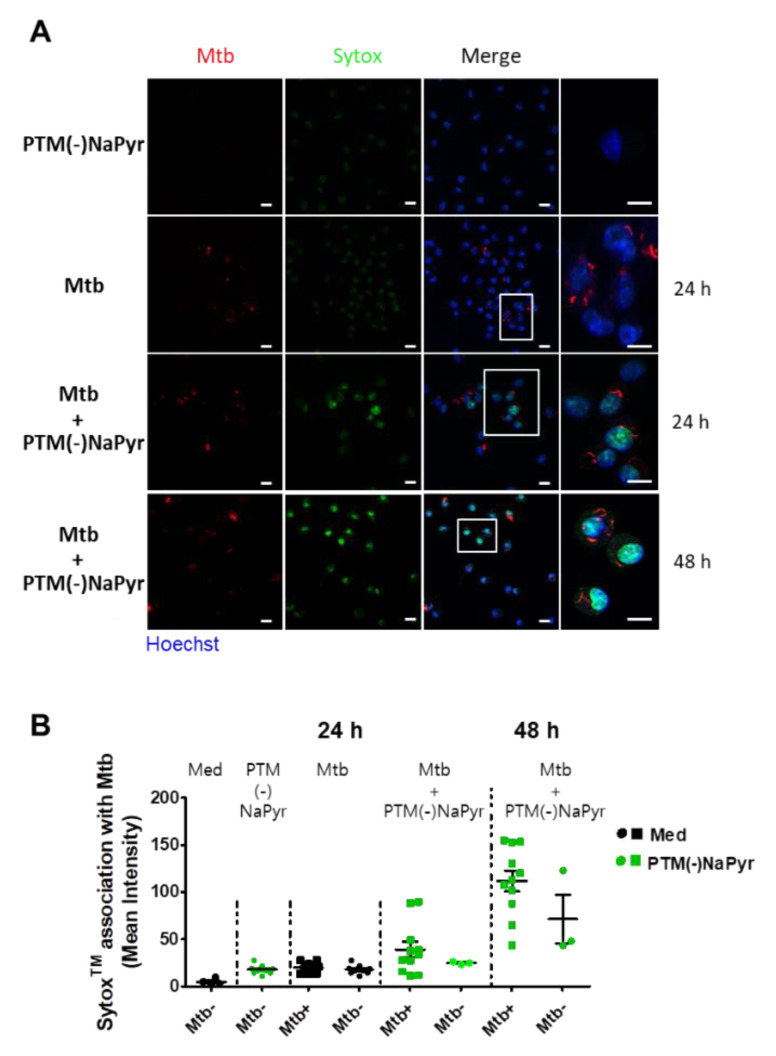
Loss of ERFP *Mtb*-infected primary macrophage membrane caused by PTM. (**A**) Association between BMDMs and SYTOx™ in uninfected cells incubated for 24 h in 8-fold-diluted PTM without NaPyr, *Mtb*-infected cells (MOI 1), and ERPF *Mtb*-infected cells incubated in 8-fold-diluted PTM without NaPyr. Last image is *Mtb*-infected cells incubated for 48 h in PTM without NaPyr. Images show stained nuclei (blue), ERFP *Mtb* (red), and SYTOx™ (green). (Right panels) Magnification highlights the association between SYTOx™ and ERFP *Mtb*-infected cells. Scale bar, 10 µm. (**B**) Quantitation of ERFP *Mtb* and SYTOx™ association in macrophages. Numbers of bacteria analyzed for each condition and proportions of *Mtb* that are considered positive for SYTOx™ association are shown. Mean fluorescence intensity was quantified using LAS X 3.7. Data are means ± SEM of one representative experiment (n = 3).

**Figure 8 biomedicines-10-01243-f008:**
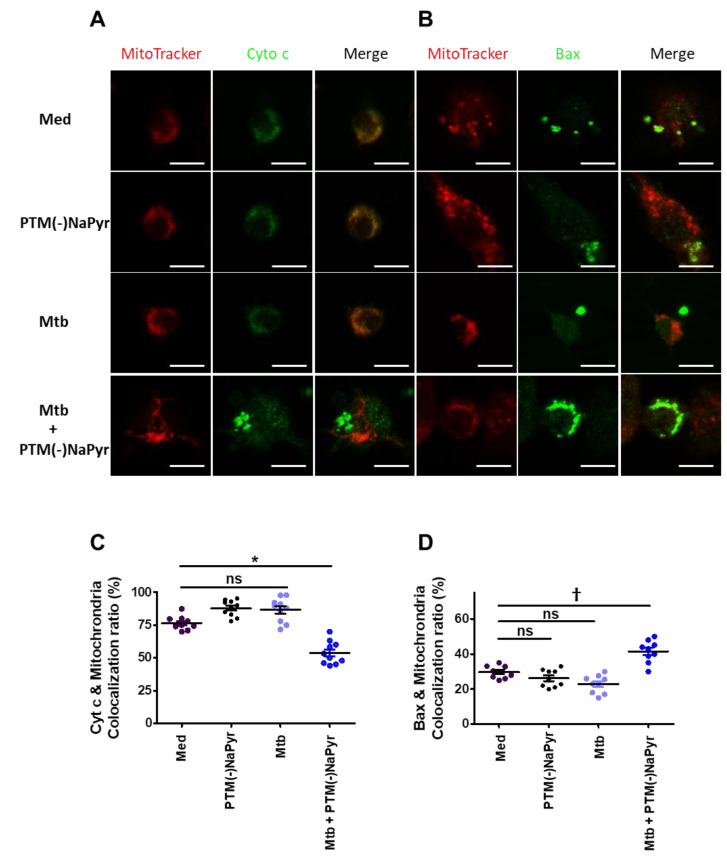
Changes in Cyt c and Bax localization in *Mtb*-infected cells incubated in PTM. (**A**,**B**) Representative confocal microscopy images show MitoTracker (red) and Cyt c or Bax (green) staining of cells incubated only in medium (Med), uninfected BMDMs incubated in PTM without NaPyr, cells infected with *Mtb*, and *Mtb*-infected cells incubated in 8-fold-diluted PTM without NaPyr for 18 h. (**A**) Cyt c, (**B**) Bax. Scale bar = 10 µm. (**C**,**D**) Quantitative co-localization of Cyt c and Bax expressed as mitochondria (n) per cell. * *p* < 0.05 and † *p* < 0.01; ns: not significant; n = 3.

**Figure 9 biomedicines-10-01243-f009:**
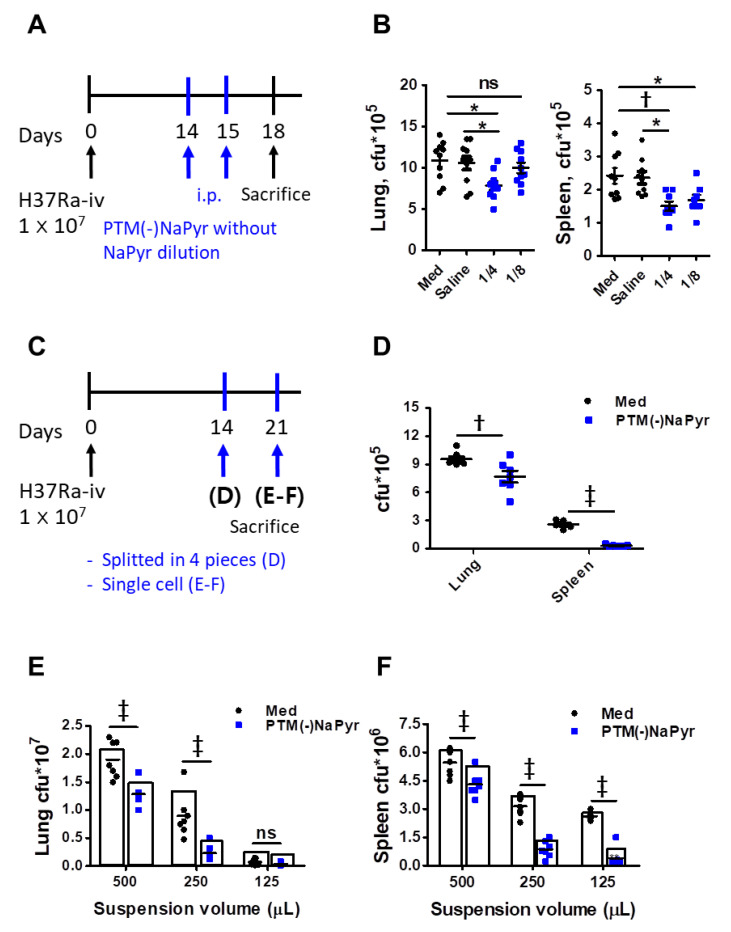
Intraperitoneal injection and direct contact with PTM reduce intracellular bacterial growth in *Mtb* H37Ra-infected mice. (**A**) Experimental schedule of PTM injected i.p. into mice *in vivo*. (**B**) Numbers of intracellular bacterial colony-forming units (CFUs) in lung and spleen tissues. (**C**) Experimental schedule of direct PTM exposure of excised spleens and lungs. (**D**) Spleen and lung tissues were cut into four sections each and incubated in 4-fold-diluted PTM without NaPyr for 1 h, then bacterial CFUs were counted. (**E**) Lungs and (**F**) spleens from *Mtb* H37Ra-infected mice were homogenized and divided into groups incubated with untreated media and 4-fold-diluted PTM without NaPyr for 1 h. All data are presented as means ± SEM of at least three independent experiments (n = 5–8 mice per group; * *p* < 0.05, † *p* < 0.01, and ‡ *p* < 0.001; ns: not significant vs. untreated mice).

## Data Availability

Data are available on request from the authors. The data that support the findings of this study are available from the corresponding author upon reasonable request.

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
