# Peer review of "Non-Thermal Plasma Jet-Treated Medium Induces Selective Cytotoxicity against Mycobacterium tuberculosis-Infected Macrophages"

_biomedicines, 2022, doi:10.3390/biomedicines10061243_

Round 1

Reviewer 1 Report

In this manuscript, the authors investigated the selective effects of non-thermal plasma jet-treated medium on cell death, Mtb intracellular replication, etc. Generally, the findings presented here are interesting, yet major improvement are required to substantiate the conclusion.

Major points:

  1. Line 13, the authors hypothesized that PTM "selectively" kills Mtb-infected cells. What is the basis that allows the authors speculate PTM "selectively" kills Mtb-infected cells other than non-infected cells?
  2. Fig. 3, it seems that PTM induces more cell death of Mtb-infected cells, only at 8-fold dilution, but not at 4-fold dilution. It is not convincing that the authors conclude the selective effects of PTM on Mtb-infected cells, considering such small dose window. Furthermore, it is not surprising that Mtb-infected cells are more susceptible to PTM given the fact that Mtb infection induces cell death.
  3. The authors should elucidate the types of cell death (apoptosis, necrosis or pyroptosis) induced by PTM, although they have provide evidence on Cytochrome C release and plasma membrane permeability.
  4. what confers the selective effects of PTM on Mtb-infected macrophages? Is it dependent on Mtb virulence or just ROS production?
  5. In the in vivo model, the authors employed the avirulent strain-H37Ra, which is unable to cause disease in experimental animals. Therefore, whether PTM could be applied to tuberculosis treatment is highly doubtful.

Minor points:

  1. Line 302, the text  (50% vs 9.06%) does not correspond to Fig. 3C. In Fig. 3C, it looks like 90% vs 60%. Fig. 3D has the same issue.
  2. Some of the figures were not properly introduced in the manuscript, such as Fig. 7.
  3. The English needs to be improved to allow for publication.

Author Response

May 14, 2022

Manuscript ID: 1686140

Dear Reviewer 1,

My colleagues and I greatly appreciate you and the reviewers for your insightful suggestions and comments. In addition, I revised one Figure in the manuscript and added Figure related to cell death mechanism in the supplementary materials. The manuscript has been carefully rechecked and appropriate changes have been made in accordance with the reviewers’ comments. I used the Track Changes mode or highlighted in Microsoft Word in the manuscript.

The responses to their comments have been prepared and attached herewith. Our responses are typed in blue.

“Please see the attachment”

I look forward to hearing from you.

Sincerely, 

Hwa-Jung Kim, MD and Ph.D.

E-mail: hjukim@cnu.ac.kr

Reviewer 2 Report

Non-thermal plasma jet-treated medium (PTM) is an upcoming therapy that is very effective against cancer cells and is a known immunomodulator.
The application of this technology is particularly interesting in the case of drug-resistant M.tb as Isoniazid resistance is globally reaching beyond 20% of total prevalence, a significant fraction of which is due to the mutation in inhA gene. Mutations in this gene make this pathogen catalase-negative and Plasma treatment has peroxides as major therapeutic mediators. 

Authors need to extensively revise English language by a native speaker with a scientific background. Phrases such as e.g.1. ". In vivo, Mycobacterium tuberculosis H37Ra were infected for 2–3 weeks and then treated" make no sense. It should be Mice were infected by Mycobacterium tuberculosis H37Ra. e.g.2. "For counting the intracellular bacteria, the cells were washed once with pre-warmed PBS and lysed in sterilized with water by means of a 30 min incubation" here it reads water was used sterilize cells!!. There are numerous examples in the manuscript making it annoyingly difficult to read. The ionic charge on nitrite ion is in subscript which is not a proper notation. Proper notation for DMEM  without Sodium Pyruvate should be DMEM(-)NaPyr... DMEM-NaPyr looks the opposite... Some places have NaPyr as a superscript, authors need to choose a uniform annotation. Mtb is the short form of a scientific name of an organism and therefore needs to be italicized throughout the script. 

Authors need to elaborate on how in vivo infections were established and their success monitored with avirulent H37Ra strain. It appears to be more of a persistence model.

Authors only showed the vulnerability of Infected macrophages to PTM. It's not clear such host cell death will enhance or control the infection as the released viable Mtb might infect many other new macrophages enhancing the infection. There is a missing aspect of pathogen survival through the ordeal of infected host cell death. The animal model is not reliable due to using the persistence model which is obligately dependent on the survival of infected macrophages and avirulent or attenuated Mtb strains such as Mtb H37Ra have been shown to induce more macrophage apoptosis when compared to virulent strains such as Mtb H37Rv. This experiment needs to monitor Mtb loads to show there is a definitive reduction. It's well documented that Macrophage apoptosis is ROS mediated and is followed by DC engulfment and facilitates the degradation of the Mtb or Mtb product containing apoptotic bodies by cross-presenting the bacterial antigens to CD8+ T cells through a process called efferocytosis. But, Superoxide and other ROS are highly potent and can directly kill intracellular microbes.

Also, the authors need to establish the mechanism of infected cell death by PTM... If it's Necrosis then it should enhance the infection. If it's autophagy and is killing Mtb has a relevant therapeutic implication.

Author Response

May 14, 2022

Manuscript ID: 1686140

Dear Reviewer 2,

My colleagues and I greatly appreciate you and the reviewers for your insightful suggestions and comments. In addition, I revised one Figure in the manuscript and added Figure related to cell death mechanism in the supplementary materials. The manuscript has been carefully rechecked and appropriate changes have been made in accordance with the reviewers’ comments. I used the Track Changes mode or highlighted in Microsoft Word in the manuscript.

The responses to their comments have been prepared and attached herewith. Our responses are typed in blue.

“Please see the attachment”

I look forward to hearing from you.

Sincerely, 

Hwa-Jung Kim, MD and Ph.D.

E-mail: hjukim@cnu.ac.kr

Round 2

Reviewer 2 Report

I can understand the limitations of the study owing to a lack of appropriate animal experiment biosafety level facilities for virulent tuberculosis models. The authors have satisfactorily defended the experiment design with appropriate references. Though these limitations of the results could have been added in discussion sections and toning down the overall claims. 

I leave this to the editor...